# Crystal structures of AztD provide mechanistic insights into direct zinc transfer between proteins

Durga Prasad Neupane[1], Stephanie Hope Fullam[1], Kelly Natalia Chacón [2] & Erik Thomas Yukl [1]

Zinc acquisition from limited environments is critical for bacterial survival and pathogenesis. AztD has been identified as a periplasmic or cell surface zinc-binding protein in numerous bacterial species. In *Paracoccus denitrificans*, AztD can transfer zinc directly to AztC, the solute binding protein for a zinc-specific ATP-binding cassette transporter system, suggesting a role in zinc acquisition and homeostasis. Here, we present the first cry
stal structures of AztD from *P. denitrificans* and tbe human pathogen *Citrobacter koseri*, revealing a beta-propeller fold and two high-affinity zinc-binding sites that are highly conserved among AztD homologs. These structures combined with transfer assays using WT and mutant proteins provide rare insight into the mechanism of direct zinc transfer from one protein to another. Given the importance of zinc import to bacterial pathogenesis, these insights may prove valuable to the development of zinc transfer inhibitors as antibiotics.

[1] Department of Chemistry and Biochemistry, New Mexico State University, Las Cruces, NM 88003, USA. [2] Department of Chemistry, Reed College, Portland, OR 97202, USA. Correspondence and requests for materials should be addressed to E.T.Y. (email: etyukl@nmsu.edu)

The mechanisms governing bacterial zinc homeostasis have garnered considerable interest in recent years owing to the importance of this element at the interface between mammalian host and bacterial pathogen[1]. In a process termed nutritional immunity, the host employs strategies to chelate available zinc in an attempt to starve invading bacteria of this essential nutrient[2,3]. Alternatively, intracellular pathogens may be exposed to toxic levels of zinc as part of the immune defense[4,5]. Thus, the ability of bacteria to maintain appropriate intracellular zinc levels in environments differing dramatically in available zinc is critical to pathogenesis.

Many pathogenic and non-pathogenic species respond to severe zinc limitation with increased expression of high-affinity ATP-binding cassette (ABC) transporters under the control of zinc-dependent transcriptional repressors such as Zur and AdcR[6]. Bacterial ABC transporters are composed of a dimeric membrane permease and ATPase as well as a periplasmic (Gram-negative) or cell surface (Gram-positive) solute-binding protein (SBP)[7,8]. The SBP confers high affinity and specificity to the system, and bacterial ABC transporters have been grouped according to the structure and substrate specificity of their SBPs[9]. By this designation, cluster A-I SBPs are specific for uncomplexed zinc, manganese, or iron. Consistent with a role in zinc acquisition from severely limited environments, deletion of zinc-specific cluster A-I SBPs results in inhibited growth in zinc-depleted media and decreased virulence in animal models[10]. Inhibition of zinc binding or transfer from these proteins thus represents a promising target for the development of novel antimicrobials. Indeed, a small molecule targeting the *Salmonella enterica* zinc SBP ZnuA inhibited growth of this organism[11]. Although the potency of this compound was insufficient for clinical use, this work nevertheless provides precedence for targeting bacterial zinc import as a means of treating infection.

It has also become increasingly clear in both Gram-negative and Gram-positive organisms that other zinc-binding proteins in the periplasm or on the cell surface, respectively, can contribute to zinc import through ABC transporters. The Zur/AdcR-regulated protein ZinT is probably the best characterized. It is fused to a ZnuA-like domain in the AdcA protein from several *Streptococcal* species[12] where it facilitates high-affinity zinc binding and rapid transfer[13,14]. In some Gram-negative organisms, the standalone periplasmic ZinT physically interacts with ZnuA and contributes to zinc import in limited conditions[13,15–18]. Crystal structures of ZinT and ZnuA along with small angle X-ray scattering data of their complex suggest a mechanism for zinc transfer from ZinT to ZnuA that depends on a flexible, His-rich loop on ZnuA, a common feature among zinc-specific cluster A-I SBPs[19]. However, although interaction between these two proteins has been conclusively demonstrated, metal transfer has not. Similarly, the polyhistidine triad protein PhtD in *Streptococcus pneumoniae* has been implicated in virulence[20,21]. Nuclear magnetic resonance experiments have further shown that the N-terminal Pht domain is able to transfer zinc to the SBP AdcAII in vitro[22], although the mechanism is not defined.

We have recently identified a periplasmic protein AztD in *Paracoccus denitrificans*, which is able to transfer zinc to the cluster A-I SBP AztC[23]. The *aztD* gene is part of the *aztABC* transporter operon and is under transcriptional control of Zur[24] as is a second zinc ABC transporter operon *znuABC*. Transfer proceeds through an associative mechanism and requires the relatively short flexible loop of AztC and each of its three conserved His residues[25]. However, unlike ZinT/ZnuA, AztC and AztD do not form a stable complex, dissociating after zinc is transferred. Further, the AztD/AztC interaction is highly specific as zinc is not transferred to *P. denitrificans* ZnuA[26]. AztD is conserved across a large number of bacterial species including human pathogens and has no homology to other putative metallochaperones. Although knockout studies in *P. denitrificans* indicated that AztD was not critical for growth in zinc limited media, they did suggest a function for this protein in zinc accumulation within the periplasm[26].

Here, we describe crystal structures of AztD homologs from *P. denitrificans* (*Pd*AztD) and *Citrobacter koseri* (*Ck*AztD), a pathogen of the carbapenem-resistant *Enterobacteriacea* group. Two high-affinity zinc-binding sites are identified, only one of which is competent for transfer to AztC. Docking studies using the previously determined structure of *P. denitrificans* AztC[25] combined with a fluorescence-based assay of transfer kinetics suggest a possible zinc transfer mechanism. To our knowledge, this work presents the first crystal structures for a new family of extracellular zinc metallochaperones and provides molecular level insights into how these proteins may participate in zinc management.

## Results

**Phylogenetic analysis of AztD**. A BLASTP search of the UniProtKB database using the protein sequence of *Pd*AztD identified 577 sequences with $E$ values below $10^{-20}$ from various bacterial taxa (Supplementary Table 1). The bulk of sequences are found in *Alphaproteobacteria* and *Gammaproteobacteria*, which include *P. denitrificans* and *C. koseri*, respectively. In the former class, the *Rhizobiales* are particularly prominent including plant symbionts and pathogens such as *Sinorhizobium meliloti* and *Agrobacterium tumefaciens*, respectively. The *Gammaproteobacteria* are dominated by *Enterobacterales*, including the human pathogen *Klebsiella pneumoniae* and its close relative *C. koseri*. N-terminal signal sequences are identified in many AztD homologs including those in the Gram-positive *Actinobacteria*, indicating that AztD homologues are predominantly periplasmic in Gram-negative species and extracellular lipoproteins in Gram-positive species. Multiple sequence alignments between representatives of the most abundant taxa indicate a large degree of divergence in AztD sequences, yet certain residues are very highly conserved (Supplementary Fig. 1). These are discussed further within the context of the crystal structures. A sequence similarity network (Fig. 1a) shows how AztD sequences cluster according to phylogenetic relationships. This network was further analyzed to determine the conservation of genome neighborhoods within and between different clusters (Fig. 1b and Supplementary Fig. 2). Overall, nearly half of all AztD sequences appear to be associated with a cluster A-I SBP (ZnuA Pfam family). Other ABC transporter genes are also highly prevalent, as are genes of the ferric uptake regulator family (Fur), which includes Zur. Finally, the CobW-CobW_C family is often associated with AztD, and a subgroup of this family (COG0523) has been linked to zinc homeostasis in a number of organisms[1,27]. Thus, the prevailing function for the AztD family would appear to be in zinc import and homeostasis. Gene neighborhood networks were also constructed independently for the five largest clusters in Fig. 1a (Supplementary Fig. 2). Each exhibits a similar pattern with the exception of cluster 4 where the only obvious link to zinc homeostasis is a prevalence of CobW-CobW_C family members. Cluster 4 is dominated by members of the *Bosea* genus of *Bradyrhizobia*, suggesting that AztD may have evolved new functions within this group. The genome neighborhood diagrams for all AztD sequences have been included as Supplementary File 1, which can be visualized by uploading to the Enzyme Function Initiative—Genome Neighborhood Tool[28,29] under the View Saved Diagrams tab at https://efi.igb.illinois.edu/efi-gnt/.

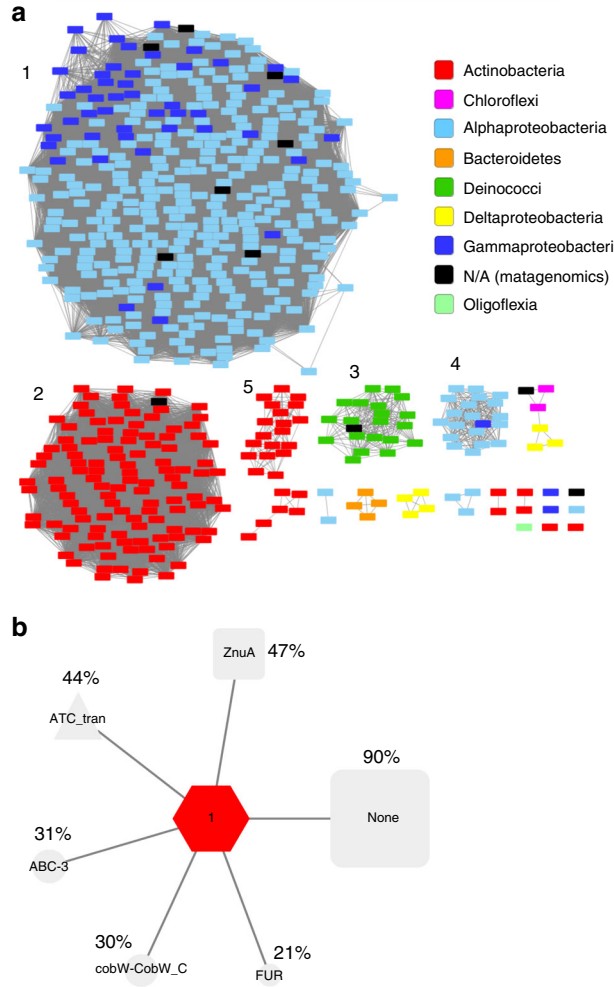

**Fig. 1** Relationships between AztD sequences. **a** Sequence similarity network including 577 sequences filtered such that only edges associated with $E$ values $< 10^{-67}$ are included in the network. Sequences are represented by rectangles colored according to class or phyla. The five largest clusters are indicated by numbers, which refer to the clusters analyzed in Fig. S2. **b** Genome neighborhood network where the red hub node represents all 577 AztD sequences. The gray spoke nodes indicate the prevalence of Pfam[72] protein family genes within 10 genes of *aztD* in at least 20% of genomes. The actual percentage of genomes with this genetic organization are indicated next to Pfam families

**Crystal structures of AztD.** Crystals of both *Ck*AztD and *Pd*AztD were grown from protein as-isolated from the periplasm of *Escherichia coli* with 0.6–0.8 equivalents of zinc. The crystal structure of *Ck*AztD was solved by single wavelength anomalous dispersion data collected on a crystal soaked with lead acetate. This structure was then used to solve the structures of native *Ck*AztD and *Pd*AztD using molecular replacement (Table 1). The crystal structure of *Ck*AztD has two monomers in the asymmetric unit while *Pd*AztD has 4. However, size exclusion chromatography indicates that both proteins exist exclusively as monomers in solution (Supplementary Fig. 3).

*Ck*AztD and *Pd*AztD share 37% sequence identity and 71% similarity, and their structures are highly similar with RMSD = 0.91–1.16 Å across backbone atoms depending on which chains are aligned (Supplementary Fig. 4). In each case, the protein adopts a beta-propeller fold composed of continuous antiparallel beta sheets forming 8, 3-, and 4-stranded blades and containing a

disulfide bond between highly conserved Cys214 and Cys231 (*P. denitrificans* numbering). A hydrated channel runs all the way through the molecule, a distance of ~25 Å with a minimum diameter of ~5.5 Å (Fig. 2). Beta-propellers are common folds in all kingdoms of life, and a DALI domain search[30,31] identifies numerous proteins containing similar domains (Supplementary File 2). Of note are the denitrification enzymes $cd_1$ nitrite reductase[32] and nitrous oxide reductase[33] as well as quinoprotein amine dehydrogenases[34–36], all of which are periplasmic enzymes encoded in the *P. denitrificans* genome. Eukaryotic proteins of the WD40 family were also identified. To date, no WD40 proteins exhibit enzymatic activity, but they are involved in a vast array of protein interaction and scaffolding functions[37]. Although nitrite reductase and nitrous oxide reductase bind heme and CuZ cluster, respectively, they have very low sequence identity to *Pd*AztD (12.3% and 11.4%) and metal-binding residues are not conserved. Further, neither is known to participate in any type of metal transfer. Thus, AztD is the first beta-propeller protein to our knowledge to have a role in metal transfer, highlighting and expanding the diversity of functions for which this fold has evolved.

Anomalous difference maps identify two distinct zinc-binding sites in *Pd*AztD (Fig. 2, zinc-ligand bond distances are listed in Supplementary Table 2). Site 1 is present in the central pore of the protein and zinc is coordinated by His124, His167, His218, and a solvent molecule. Site 2 is contained on an extension from the body of the protein composed of the N- and C-terminal beta strands and a beta hairpin from blade 2. Zinc is coordinated by His99, His102, Asp104, and the side chain of the C-terminal His408. Zinc ligands at both sites are highly conserved in ≥95% of AztD homologues with the exception of His408, which is only conserved in 68.1% of analyzed sequences (Supplementary Fig. 1). Gly100, which is part of the beta turn at site 2, is also highly conserved at 93.4%.

These zinc sites are occupied in the holo *Ck*AztD structure as well (Fig. 3, zinc-ligand bond distances are listed in Supplementary Table 3). All zinc occupancies refined to > 0.7 in *Pd*AztD and 0.4 in *Ck*AztD. However, site 2 zinc ions in certain chains exhibited very weak anomalous difference density and high B factors indicative of low occupancy (Supplementary Table 4). Part of this variability likely arises from crystal-packing effects, particularly in the *Ck*AztD structure. Poorly occupied sites also exhibited weak 2Fo−Fc density for ligand residues, indicative of disorder when zinc is absent (Supplementary Fig. 5). Indeed, although the crystal structures of apo and holo *Ck*AztD are very similar overall (RMSD = 0.22–0.44 Å) (Fig. 3), residues 99–104, including zinc ligands His101 and His104, could not be modeled in three of four chains of the apo structure. In the chain where these residues could be modeled, they were displaced from their positions in the holo structure. The C terminus including His405 was disordered in all chains. In contrast, very little structural perturbation occurs for ligands to site 1 when zinc is absent.

The issue of partial zinc occupancy is perhaps unsurprising as crystals were grown from protein as-isolated with slightly substoichiometric zinc. In fact, solution binding data indicated that a third zinc ion binds to WT *Pd*AztD with relatively low affinity[23] (Table 2). There is a short, His-rich tract of residues at the N terminus that likely accounts for this (Supplementary Fig. 1). Similar tracts of residues are identified at the termini or inserted into the sequences of AztD homologues, including one at the C terminus of *Ck*AztD. In the crystal structures of both *Ck*AztD and *Pd*AztD, these regions are disordered and no anomalous difference density is observed. Thus, if these regions do bind zinc, they do so with relatively low affinity and are unoccupied in the current structures.

**Table 1 Data collection and refinement statistics**

|  | *Pd*AztD native | *Ck*AztD native | Apo *Ck*AztD | *Ck*AztD Pb derivative |
|---|---|---|---|---|
| *Data collection* |  |  |  |  |
| Space group | $P2_12_12_1$ | $P2_1$ | $P2_1$ | $P2_1$ |
| Cell dimensions |  |  |  |  |
| $a, b, c$ (Å) | 89.5, 96.4, 175.5 | 54.2, 128.8, 56.9 | 56.9, 127.9, 113.2 | 53.6, 128.1, 57.2 |
| $\alpha, \beta, \gamma$ (°) | 90.0, 90.0, 90.0 | 90.0, 105.2, 90.0 | 90.0, 94.5, 90.0 | 90.0, 100.4, 90.0 |
| Wavelength (Å) | 1.00000 | 1.00000 | 1.00000 | 0.95007 |
| Resolution (Å) | 79.74–2.15 | 64.40–1.73 | 48.4–1.98 | 48.7–2.33 |
| $R_{sym}$ or $R_{merge}$ | 0.10 (0.61)* | 0.06 (0.37) | 0.07 (0.52) | 0.07 (0.63) |
| $I/\sigma I$ | 19.7 (2.8) | 10.2 (2.7) | 9.7 (2.1) | 13.9 (2.2) |
| Completeness (%) | 96.8 (79.0) | 99.2 (98.6) | 98.6 (96.4) | 99.7 (99.8) |
| Redundancy | 7.5 (4.9) | 3.2 (3.1) | 3.4 (3.1) | 3.7 (3.5) |
| *Refinement* |  |  |  |  |
| Resolution (Å) | 79.74–2.15 | 64.4–1.73 | 48. 4–1.98 |  |
| No. of reflections | 606,009 | 246,042 | 379,354 |  |
| $R_{work}/R_{free}$ | 17.9/23.7 | 15.3/18.8 | 19.5/23.8 |  |
| No. of atoms |  |  |  |  |
| Protein | 11,126 | 6304 | 21,724 |  |
| Ligand/ion | 8 | 4 | 0 |  |
| Water | 885 | 700 | 501 |  |
| *B*-factors | 32.5 | 30.8 | 42.0 |  |
| R.m.s. deviations |  |  |  |  |
| Bond lengths (Å) | 0.009 | 0.005 | 0.013 |  |
| Bond angles (°) | 1.066 | 0.947 | 1.259 |  |

Each data set was collected from a single crystal. *Values in parentheses are for highest-resolution shell

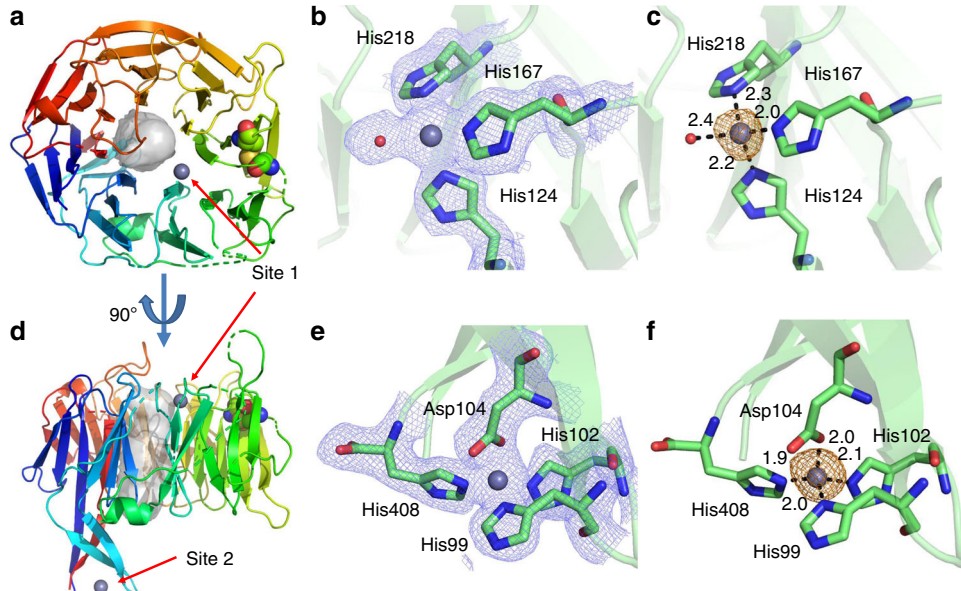

**Fig. 2** Structure of *Pd*AztD. **a** and **d** Structure is shown as cartoon colored blue to red from N- to C-terminus, zinc ions, and disulfide bonded Cys residues are shown as spheres colored according to element. A channel calculated using MoleOnline[73] is shown as gray surface. **b**, **c**, **e**, and **f** Zoomed in views of zinc-binding sites 1 and 2 with zinc shown as gray spheres at 0.5 VDW radius, side chains shown as sticks colored according to element and 2Fo–Fc density shown as blue mesh contoured at 1.0 σ **b** and **e** and anomalous difference density shown as orange mesh contoured at 5.0 σ **c** and **f**. Supplementary Table 4 lists occupancies, B-factors, and anomalous densities for all zinc sites in the asymmetric unit

**Zinc binding and transfer from WT and mutant AztD**. The crystal structures do not allow us to determine whether associative transfer of zinc to AztC occurs from *Pd*AztD site 1, site 2 or both. In order to evaluate these possibilities, each of the zinc coordinating residues from site 1 or site 2 were mutated to Ala. The mutant lacking site 1 (H124A/H167A/H218A) is called ΔS1 and that lacking site 2 (H99A/H102A/D104A/H408A) is called ΔS2. The zinc-binding properties of each mutant were

characterized by Mag-Fura 2 competitive fluorescence assay (Supplementary Fig. 6, Table 2). Each mutant binds two zinc ions as compared with three distinct binding events identified in WT AztD, indicating that one binding site is lost in each mutant as expected. The third and lowest affinity zinc-binding site is retained in each mutant, consistent with its likely location at the N terminus. Thus, each mutant has one high-affinity and one low-affinity binding site. The observation that the high-affinity

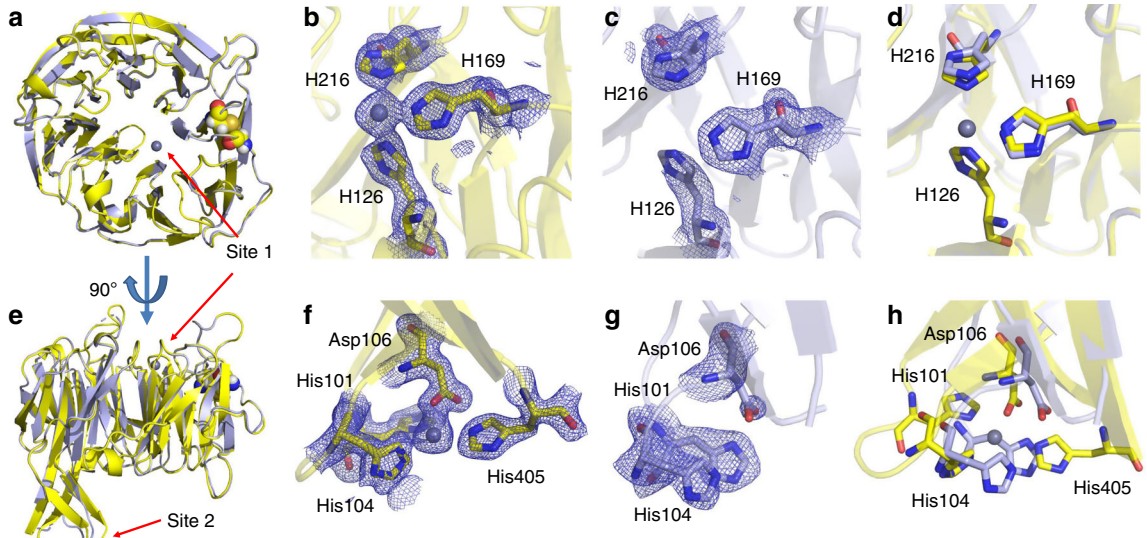

**Fig. 3** Structure of apo (blue) and holo (yellow) CkAztD. **a** and **e** Overall aligned structures shown as cartoon. Zinc ions and disulfide bonded Cys residues are shown as spheres colored according to element. **b–d** and **f–h** Zoomed in views of zinc-binding sites 1 and 2 with zinc shown as gray spheres at 0.5 VDW radius, side chains shown as sticks colored according to element and 2Fo–Fc density shown as blue mesh contoured at 1.0 σ **b**, **c**, **f**, and **g**. Overlay of zinc-binding site residues for apo and holo **d** and **h**

### Table 2 Zinc-binding affinity and stoichiometry of WT and mutant AztD

| Protein | Site | $K_d \pm$ S.D. (nM) ($n = 3$) |
|---|---|---|
| WT PdAztD[23] | 1 | 0.7 ± 0.3 |
| | 2 | 54 ± 8 |
| | 3 | 340 ± 110 |
| ΔS1 PdAztD | 2 | 1.3 ± 0.7 |
| | 3 | 248 ± 81 |
| ΔS2 PdAztD | 1 | 0.1* |
| | 3 | 158 ± 35 |
| WT PdAztC[40] | 1 | 0.3 ± 0.1 |

*Uncertainties for this value could not be estimated as the $K_d$ appears to be below the detection limit of this assay. The data were fitted with the indicated value

binding sites of both mutants exhibit $K_d$'s near the measurable limit of the assay (~0.5 nM) suggest that there may be some negative cooperativity between binding events in the WT protein. Further, the comparable binding affinities between WT or mutant AztD and AztC indicate that any efficient and directional transfer process must be under kinetic rather than thermodynamic control as has been discussed previously[23].

Zinc transfer was evaluated by an intrinsic fluorescence assay (Fig. 4) similar to that described previously[23], which relies on the large increase in fluorescence emission of AztC upon zinc binding. In this case, AztD as-isolated was fully reconstituted with two equivalents of zinc and titrated into a solution of apo AztC. Ethylenediaminetetraacetic acid (EDTA) at 1 mM was included to prevent zinc acquisition by AztC from solution or adventitious binding sites. AztD was added up to 1.6 equivalents followed by sufficient zinc to saturate both EDTA and AztC. As has been observed previously, the emission intensity at 315 nm increases with a constant slope up to approximately a single equivalent of added AztD. Subsequent additions of protein lead to much smaller increases. This is owing to the added fluorescence of AztD without zinc transfer as has been confirmed previously using control titrations of apo AztC with apo AztD under the same conditions[23]. These results indicate a 1:1 zinc transfer

stoichiometry. Saturation of the AztC zinc-binding site is confirmed by subsequent addition of saturating zinc chloride, which causes very little additional change in fluorescence intensity. The data for ΔS1 AztD is essentially identical to the WT, indicating that site 2 alone is sufficient for efficient metal transfer. In contrast, the plot of ΔS2 AztD versus fluorescence intensity exhibits a small, consistent slope throughout the titration. Further, addition of zinc chloride at the end of the titration results in a large increase, indicating that little if any zinc was transferred to AztC from this mutant.

In order to determine the fate of the second high-affinity zinc of AztD, transfer assays were performed by incubation of equal concentrations of reconstituted AztD and apo AztC in the presence of EDTA followed by chromatographic separation and zinc quantitation (Fig. 5, Supplementary Fig. 7). The results show that ~ 1 equivalent of zinc is transferred from WT and ΔS1 AztD to AztC. Any other zinc originating from AztD is apparently lost to solution, as none remains associated with AztC. Very little zinc is transferred from ΔS2 AztD to AztC and nearly all is lost to solution. Zinc binding to apo AztC in this case is similar to the control containing no AztD, indicating both that ΔS2 AztD is deficient for zinc transfer and that AztC has a very high affinity for zinc. These data are consistent with fluorescence results indicating that zinc is transferred only from site 2. Finally, reconstituted AztD in the absence of AztC loses both zinc ions during incubation with EDTA and subsequent chromatography, demonstrating that both AztD zinc sites are relatively labile. Thus, AztC acts as a kinetic trap for zinc transferred from AztD in the presence of competing chelators.

**Zinc is not transferred from site 1 to site 2 of AztD.** The above results confirm efficient, associative zinc transfer from AztD site 2. However, the function of site 1, which is also highly conserved, is not clear. In the structure of native CkAztD, the feature containing site 2 in chain A projects into the pore of chain B, making a hybrid site with two zinc ions and His104 apparently bridging between them (Supplementary Fig. 5). A buffer molecule introduced during crystallization is also nearby in the pore. This suggested the possibility of associative, kinetically controlled

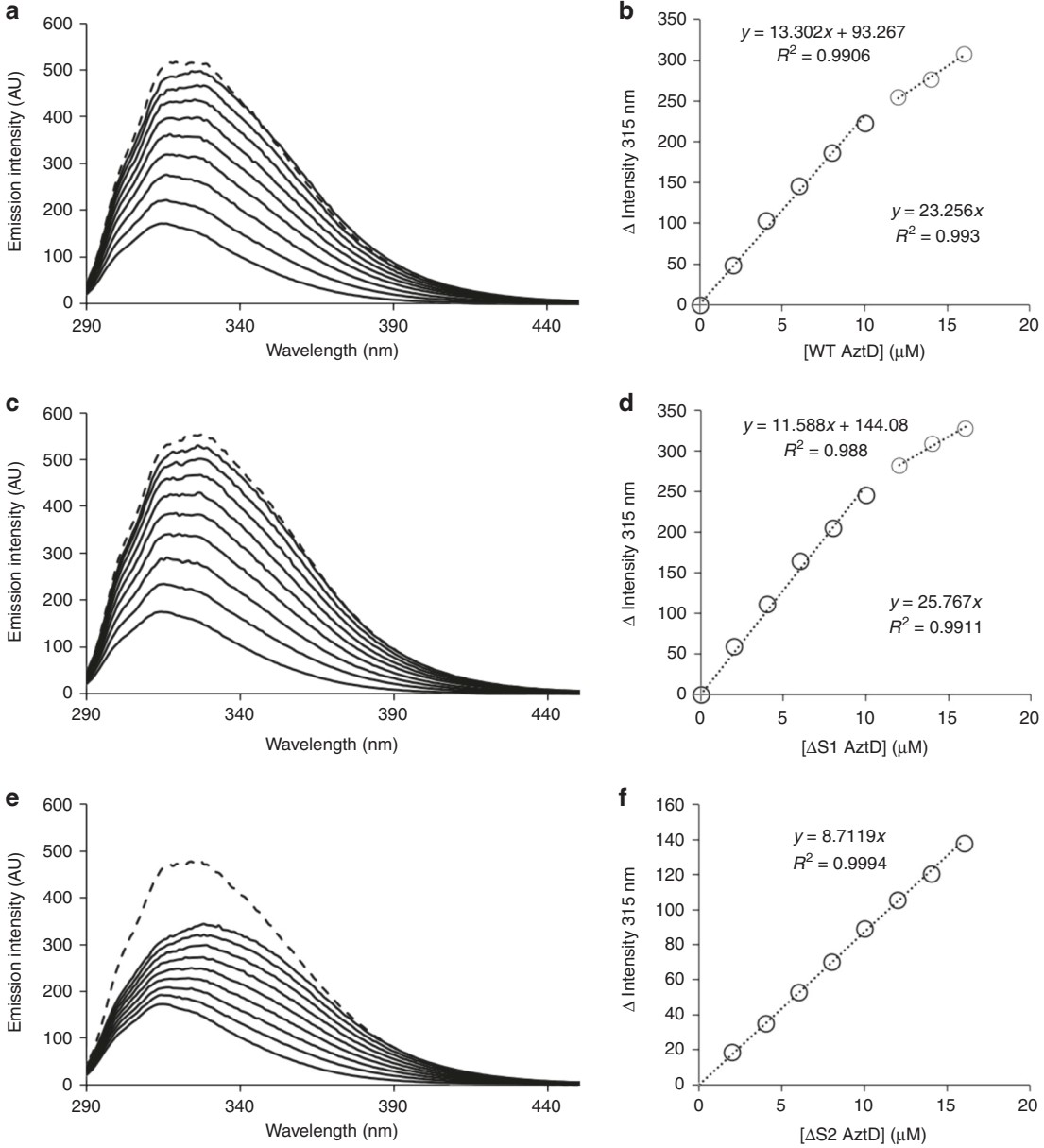

**Fig. 4** Zinc transfer by intrinsic fluorescence. Apo *Pd*AztC was titrated with reconstituted WT *Pd*AztD **a** and **b**, ΔS1 *Pd*AztD **c** and **d** or ΔS2 *Pd*AztD **e** and **f** in the presence of 1 mM EDTA. Fluorescence emission spectra **a**, **c**, and **e**; $\lambda_{exc} = 278$ nm) were recorded after each addition of *Pd*AztD and the intensity change at 315 nm plotted as a function of *Pd*AztD concentration **b**, **d**, and **f** with equations for linear fits indicated. A saturating concentration of ZnCl$_2$ (1.1 mM) was added after the titration to assess whether transfer from AztD was complete (dotted line, **a**, **c**, and **e**)

transfer from site 1 to site 2 between two AztD molecules. To test if this process can compete with dissociation of zinc and capture by EDTA, we titrated apo AztC with zinc-reconstituted ΔS2 AztD in the presence of EDTA and stoichiometric apo ΔS1 AztD (Supplementary Fig. 8). In this case, the only source of zinc is bound at site 1 of the ΔS2 *Pd*AztD, which cannot be passed directly to AztC. However, if zinc can be passed to site 2 of apo ΔS1 AztD, it could be subsequently transferred to AztC resulting in a large, saturable fluorescence increase. This does not appear to be the case, as titration results are comparable to those with ΔS2 AztD alone (Fig. 4e, f). Thus, if zinc is transferred associatively between AztD sites, this process does not compete with its dissociation to solution under these conditions, and the hybrid zinc site in the crystal structure of *Ck*AztD is most likely an artifact of crystallization.

**Zinc transfer kinetics**. The large intrinsic fluorescence increase in AztC upon zinc binding allows for this property to be used to measure the kinetics of zinc transfer. Apo AztC was rapidly mixed in a stopped-flow apparatus with an excess of reconstituted AztD to maintain pseudo-first order conditions. EDTA was again included to prevent non-specific zinc binding to AztC. At each concentration of AztD, the increase in fluorescence intensity fit well to a single exponential function (Fig. 6a). A plot of $k_{obs}$ versus AztD concentration is linear, and the slope indicates a second order rate constant for the overall process of 575 M$^{-1}$s$^{-1}$ (Fig. 6b). Thus, the transfer is quite slow, reaching completion on the seconds to minutes time scale under these conditions. Given this observation, further kinetic experiments were performed at lower concentrations in a stirred cell with a standard fluorometer, allowing us to compare the kinetics of transfer to WT and Y121A

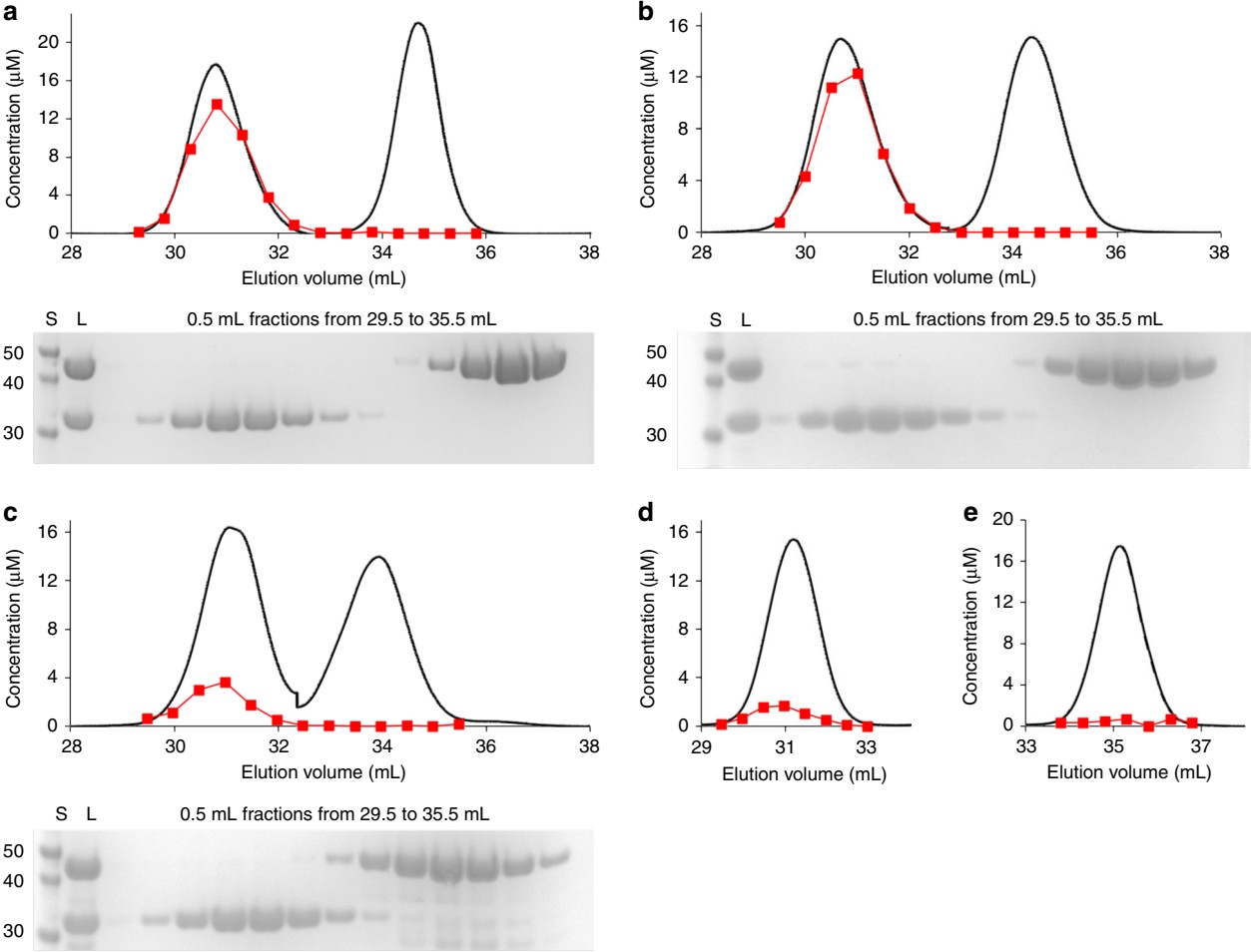

**Fig. 5** Zinc transfer by anion exchange chromatography. UV chromatograms (black lines) and zinc quantitation of chromatographic fractions (red squares) of reconstituted WT AztD **a**, ΔS1 AztD **b**, and ΔS2 AztD **c** incubated with apo AztC in the presence of 1 mM EDTA. **d** AztC or **e** AztD were incubated alone under identical conditions used for transfer assays. SDS–PAGE results are shown for **a–c** and include a MW standard (S), the AztC/AztD protein mixture loaded onto the column (L) and relevant chromatographic fractions

AztC (Fig. 6c, d). We had previously shown that this mutant is capable of acquiring zinc from AztD[25] but appeared to take longer to reach completion. Indeed, transfer to WT AztC was faster under these conditions, yielding a second order rate constant of $1330\,M^{-1}s^{-1}$ as compared with $280\,M^{-1}s^{-1}$ for the mutant. This WT rate constant is greater than that acquired by stopped-flow but comparable, particularly considering the very different concentration regimes at which the two experiments were performed (0.5 μM versus 37.5 μM AztC).

## Discussion

Examples of bacterial zinc metallochaperones in the literature are quite rare. ATPases of the COG0523 family have been linked to zinc homeostasis and may have a role as intracellular zinc chaperones[27,38,39], although their precise function remains enigmatic. As previously mentioned, ZinT and PhtD interact with zinc ABC transporter SBPs and are important for zinc homeostasis, but detailed information on the mechanism of metal transfer remains elusive. Our previous work with the *aztABCD* operon of *P. denitrificans* demonstrated that this is a zinc-specific ABC transporter system[24,40] and that AztD has a role in zinc acquisition in the periplasm[26] and is capable of associative transfer of zinc to the SBP AztC[23,25]. Here we show that AztD homologues are found in a large number of diverse bacterial species where the predominant function appears to be in zinc acquisition and homeostasis based on the genome neighborhood. Although overall sequence conservation is low, zinc-binding residues of both site 1 and site 2 as well as a number of Gly and Pro residues are highly conserved (Supplementary Fig. 1). The structures of AztD from *P. denitrificans* and *C. koseri* reveal a beta-propeller fold. The conserved Gly and Pro residues are often found at the turns between beta strands, suggesting that this fold is conserved among the AztD family. Similarly, the Cys residues engaged in a disulfide bond are highly conserved. Thus, it appears that AztD proteins are a widespread family of extracellular zinc metallochaperones.

The kinetic data presented here suggests that formation of the AztD/AztC complex is rate limiting and relatively slow. This suggests at least two possible kinetic models:

Figure 7a assumes that the fluorescence change is tracking the formation of a zinc transfer intermediate (AztD–Zn–AztC) whose fluorescence emission is very similar to that of the final products. In this case, the rate of transfer ($k_T$) could not be evaluated independently. Alternatively, a pre-transfer intermediate (Zn–AztD–AztC) may be formed in a slow step to which our fluorescence data is insensitive. This would be followed by rapid formation of the transfer complex (Fig. 7b) followed by dissociation. In either case, the kinetic data strongly suggest that the protein association phase is rate limiting in the transfer process.

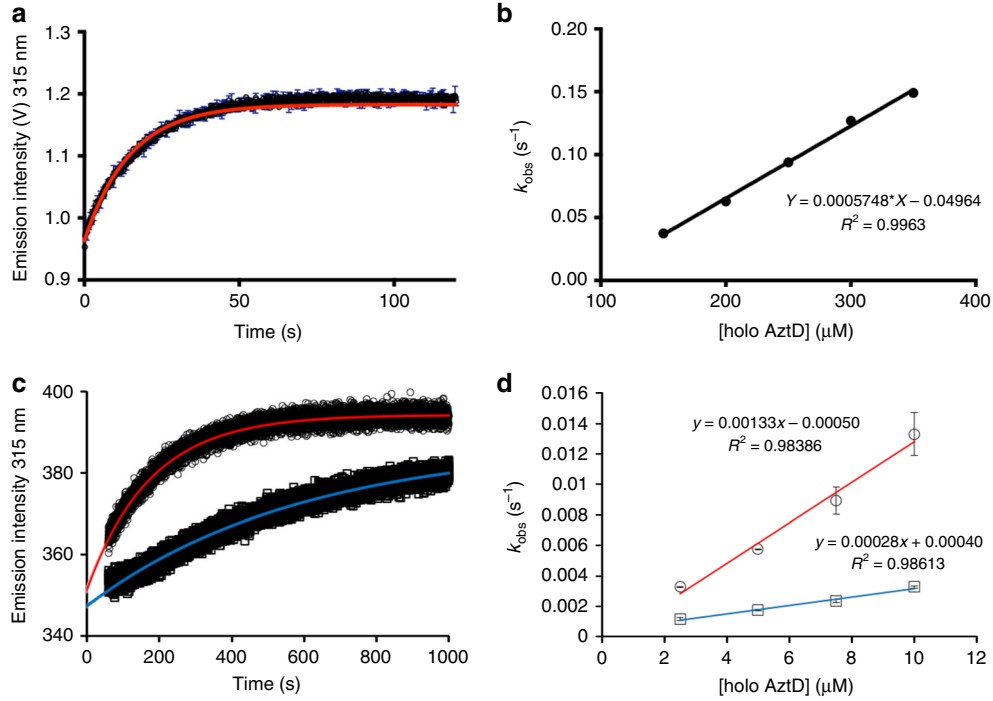

**Fig. 6** Kinetics of zinc transfer from AztD to AztC. **a** Representative stopped-flow data (black) with fit to a first order kinetic scheme (red). **b** Observed rate constant versus AztD concentration under pseudo-first order stopped-flow conditions. Error bars represent standard deviations from duplicate experiments ($n = 2$). **c** Representative kinetic data from standard fluorescence emission experiments for WT (circles) and Y121A (squares) with fits to a first order kinetic scheme (red and blue, respectively). **d** Observed first order rate constant versus AztD concentration for standard fluorescence data under pseudo-first order. Error bars represent standard deviations from triplicate experiments ($n = 3$). Symbols and linear fit colors are as for **c**

**a**
$$ZnAztD + AztC \xrightarrow[\text{Slow}]{k_{on}} AztD\text{-}Zn\text{–}AztC \xrightarrow{k_T} AztD + ZnAztC$$

**b**
$$ZnAztD + AztC \xrightarrow[\text{Slow}]{k_{on}} Zn\text{-}AztD\text{–}AztC \xrightarrow[\text{Fast}]{k_T} AztD\text{–}Zn\text{-}AztC \xrightarrow{k_D} AztD + ZnAztC$$

**Fig. 7** Kinetic models. Kinetic models for zinc transfer assuming **a** direct formation of a zinc transfer intermediate or **b** slow formation of a pre-transfer intermediate followed by rapid formation of a transfer intermediate

This is in contrast to what is known for CusCBAF, a copper exporter of the resistance-nodulation-division family[41–45]. In that system, the small periplasmic protein CusF acts as a metallochaperone, delivering $Cu^+$ or $Ag^+$ to CusB[46,47]. The mechanism involves the nearly diffusion limited formation of a CusB/CusF complex with shared ligation to the copper ion[48,49]. Whether AztD and AztC form a similar intermediate with shared ligation to the zinc is uncertain from our data. The fluorescence emission would likely report on the formation of such a complex as well as the final product of transfer (holo AztC). However, whether a pre-transfer complex is formed or a shared ligation intermediate is formed directly, it is this step that is rate limiting as demonstrated by the dependence of the transfer rate on AztD concentration.

Using the crystal structures of PdAztD and apo PdAztC (PDB ID: 5W56)[25], docking models were created showing possible structures that the AztD–AztC transient complex might adopt. Our initial attempt using the HADDOCK webserver[50] (Supplementary Fig. 9) did not show a clear role for the AztC flexible loop, which is known to be critical for transfer[25]. In order to improve the model, we considered the small angle X-ray scattering structure of the complex of ZnuA with ZinT, where the ZnuA flexible loop was proposed to be inserted into the deep

zinc-binding cleft of ZinT[15]. Thus, we began with a manually docked complex where the AztC flexible loop was inserted into the central channel of AztD. This was then refined in the refinement only module of the HADDOCK webserver. The resulting complex (Fig. 8a) was improved over the previous attempt as judged by interaction energies and HADDOCK score (Supplementary Table 5), which is a weighted sum of multiple model quality metrics. Loop insertion positions AztD site 2 directly into the AztC zinc-binding cleft, with the zinc ~ 10 Å from its final destination (Fig. 8b). The position of the sole Trp140 near the zinc-binding site likely explains the sensitivity of fluorescence emission to zinc binding. AztC loop residues His120, Tyr121, and His122 engage hydrogen bond interactions with AztD residues lining the pore (Fig. 8c). In addition, the side chain of Tyr121 is packed between AztD Pro310 and Gly362 on two adjacent loops. These interactions likely position the two proteins for zinc transfer. Although any transfer model will require experimental verification, the model presented in Fig. 8 is consistent with the existing data and suggests several interesting mechanistic elements as discussed below.

The kinetic data are consistent with the proposed transfer model, which could be thought of as a pre-transfer intermediate. The AztC loop is highly flexible, owing to Gly residues at positions 117–119, 126, and 133. Thus, many conformations of this feature must be sampled before a productive complex with AztD can be formed. Further, intermolecular interactions are quite extensive, in contrast to the in silico models predicted for the CusB-Cu-CusF complex[49] whose formation is much faster[48]. The model also justifies the observation that transfer to the Y121A mutant is slower than WT as the side chain of this residue packs between AztD residues and engages a hydrogen bond. In the docked orientation, the zinc-binding sites of the two proteins are mirror images of one another. It is tempting to speculate that a shared ligation intermediate would have purely His coordination,

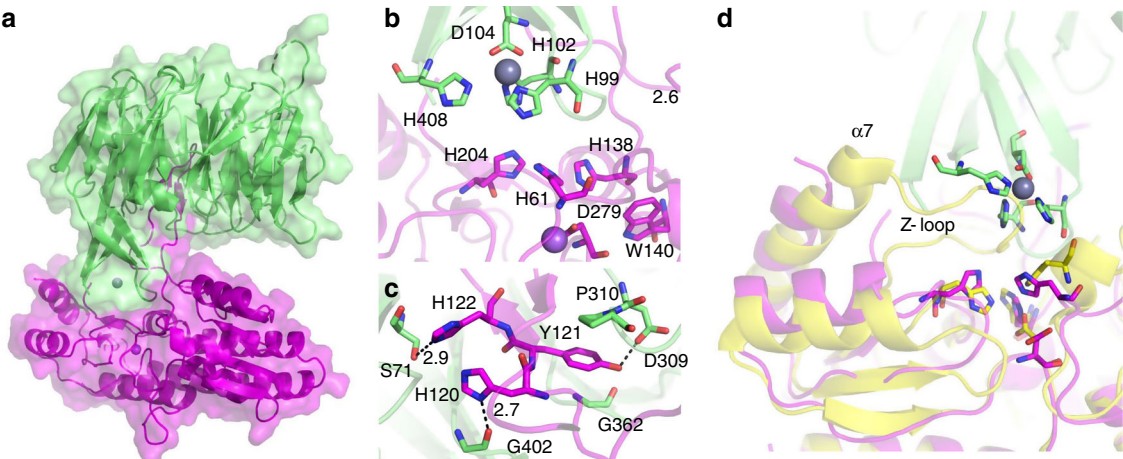

**Fig. 8** Docking model of apo *Pd*AztC (magenta) and *Pd*AztD (green). **a** Intact complex showing surface and secondary structure of each protein, zinc is shown as a gray sphere and sodium shown as a purple sphere. Zoomed region of the complex showing the positions of **b** zinc-binding residues and **c** interactions mediated by important loop residues of AztC. Side chains are shown as sticks colored according to element, zinc is shown as a gray sphere and sodium shown as a purple sphere. **d** Holo AztC (yellow, PDB ID: 5W57) aligned with apo AztC in the AztC/AztD complex showing the position of relevant features. Side chains are shown as sticks colored according to element, zinc is shown as a gray sphere. The sodium ion of apo AztC is omitted for clarity

either by all His ligands in a six-coordinate complex or by a subset to generate the more common tetrahedral geometry[51]. Zinc has been observed to cause dimerization of His-tagged proteins, as precedence for the formation of shared zinc sites with His ligation[52]. The final step in transfer would then be expected to be displacement of the remaining AztD ligands by binding of AztC Asp279 to the zinc. Concomitant dissociation of the sodium ion coordinated by Asp279 in the apo form has been suggested to drive conformational changes associated with the holo form, namely closure of the α7 helix and so-called Z-loop over the zinc-binding site[25] (Fig. 8d). This may represent a mechanism whereby *Pd*AztC can disengage from *Pd*AztD after zinc transfer occurs, as these motions would cause steric clashes with AztD. Further experiments including analyzing the effect of pore blocking molecules as transfer inhibitors and independently characterizing protein–protein association rates using Förster resonance energy transfer or surface plasmon resonance will be important to validate the above model. Nevertheless, it provides an excellent starting point for these studies, which are underway in our laboratories.

The mutagenesis data clearly demonstrates that zinc is transferred exclusively from AztD site 2 to AztC (Figs. 4 and 5), leaving one to question the function of the highly conserved site 1. Comparison of zinc-binding affinities of WT and mutant AztD proteins (Table 2) suggests some negative cooperativity. The two high-affinity binding events in WT exhibit $K$d's of 0.7 and 54 nM, whereas each of the ΔS1 and ΔS2 mutants exhibit a single high-affinity site with $K$d ≤ ~1 nM. The cooperative nature of binding between sites 1 and 2 suggest that site 1 may function to tune the overall zinc-binding affinity to the needs of the organism. Alternatively, its high-affinity may enable a function in zinc storage, whereas a rapid off-rate would allow non-specific dissociative transfer to other zinc-binding proteins of the periplasm. Although both zinc sites are labile as demonstrated by loss of zinc to EDTA during chromatography (Fig. 5e), zinc bound at site 2 can be efficiently transferred to AztC, which serves as a kinetic trap.

In conclusion, this work provides high-resolution structures for a newly discovered family of extracellular zinc metallochaperones. Combined with kinetic data, these structures allow us to construct a testable model for the mechanism of zinc transfer. Given the importance of zinc acquisition and homeostasis to pathogenesis

and the conservation of AztC and AztD in human pathogens, such a model may prove valuable for the development of novel antibiotics that target zinc binding and transfer.

## Methods

**Phylogenetic analysis of AztD**. The protein sequence of *Pd*AztD (UniProtKB accession number A1B2F4) was used to perform a BLAST search of the UniProtKB database[53]. Sequences were filtered to include only those with $E$ values below $10^{-20}$. These sequences were submitted to the Enzyme Function Initiative—Enzyme Similarity Tool[28,29,54] to generate a sequence similarity network, which was further processed and visualized in Cytoscape[55]. Genome neighborhood networks were generated by Enzyme Function Initiative—Genome Neighborhood Tool. Sequences were aligned using Clustal Omega[56] and conservation of residues analyzed in Jalview[57].

**Expression, purification, and characterization of proteins**. Expression, purification, quantitation, and apo protein generation of *Pd*AztC and *Pd*AztD from *P. denitrificans* PD1222 were performed as previously described[23,40]. In brief, proteins were overexpressed in *E. coli*, extracted from the periplasm using an osmotic shock protocol and purified by anion exchange and size exclusion chromatography. Proteins were quantified using absorbance at 280 nm and apo proteins were generated by extensive dialysis against EDTA at low pH. Mutants of *Pd*AztD were generated using the Q5® site-directed mutagenesis kit (New England Biolabs) and confirmed by sequencing. The full-length *aztD* gene from *C. koseri* strain 4225–83 was amplified by PCR from genomic DNA (ATCC® BAA-895D) using the following primers: 5′-ACTATCATATGATGGAGAACATTATGAA-GAAACG-3′ Fwd and 5′- ACTATGGTACCTTAATGAGCGTGATCATCA-3′ Rev. The PCR product was cloned into a pCDFDuet vector (Novagen) using *Nde*I and Acc651 restriction sites (underlined) and transformed into BL21 *E. coli* cells. Procedures for expression and purification were identical to those of used for the homologous *P. denitrificans* protein[23] except that ion exchange chromatography was carried out entirely at pH 8.0. An extinction coefficient of 28,545 M$^{-1}$ cm$^{-1}$ was determined for *Ck*AztD using the method of Edelhoch[58] and was used to quantify protein. Apo proteins were generated in all cases according to the previously described dialysis method[40] and confirmed to contain < 0.05 equivalents zinc. For zinc quantitation, 200 μL of protein at 25 μM was digested in 4 M HNO$_3$ at 70 °C overnight and diluted to 2.5 mL prior to triplicate analysis using a Perkin-Elmer 2100 DV inductively coupled plasma–optical emission spectrometer (ICP–OES), calibrated with a multielement standard (Alpha Aesar) at a wavelength of 213.857 nm.

**Crystallization, structure determination, and docking**. Initial crystallization hits were identified using the Hauptman–Woodward Institute standard screen[59] and subsequently optimized to generate diffraction quality crystals. *Pd*AztD crystals were grown by hanging drop vapor diffusion in VDX plates (Hampton Research). Drops contained a 1:1 ratio of protein at 10 mg/mL with reservoir solution containing 20–26% PEG 4000, 0.1 M acetate buffer pH 5.4, 0.1 M magnesium nitrate and 2.5% w/v 1-ethyl-3-methylimidazolium chloride (Ionic liquid screen #9, Hampton Research). Crystals of *Ck*AztD in both as-isolated and apo forms were

grown under paraffin oil using a 1:1 ratio of 10 mg/mL CkAztD and precipitant solution containing 20–26% PEG 4000, 0.1 M MES buffer pH 6.0 and 0.1 M sodium nitrate or 6% v/v Tacsimate pH 7.0. To generate lead derivatives, crystals were soaked in mother liquor lacking tacsimate and containing 10 mM lead acetate for 10 min, followed by backsoaking and cryoprotection for ~1 min in mother liquor containing 10% PEG 400 prior to cryocooling in liquid nitrogen.

Diffraction data were collected at 100 K and 1.00000 or 0.95007 Å on beamlines 8.2.2 and 5.0.2 at the Advanced Light Source at Berkeley National Laboratory. Data sets were indexed and integrated with XDS[60,61] and scaled using Aimless[62]. The structure of CkAztD was solved using single wavelength anomalous dispersion data from a lead derivative crystal using Phenix AutoSol[63]. The initial solution was subjected to several rounds of automated model building, density modification, and refinement using the Phenix AutoBuild Wizard[64]. The semi-refined structure was used as the search model for molecular replacement using Phaser-MR[65] for both native and apo CkAztD and native PdAztD. Manual model building was done in Coot[66]. Refinement and calculation of anomalous difference maps were performed using the Phenix suite[67]. The final coordinates of native and apo CkAztD and native PdAztD have been deposited in the PDB with entry codes 6CMK, 6N01, and 6CK1, respectively. Structures contained 0.3%, 0.1%, and 0.2% Ramachandran outliers, respectively.

Docking was performed using the HADDOCK[68] webserver[50]. The structures of the A-chain of apo PdAztC (PDB ID: 5W56) and the D-chain of native PdAztD were either submitted as-is with potential interacting residues listed for complete docking, or a manually docked complex of these proteins was submitted for refinement only. Figures were prepared using Pymol (http://www.pymol.org), which was also used for pairwise structural alignments.

**Fluorescence spectroscopy.** Zinc-binding affinities were measured using a Mag-Fura 2 competition assay derived from Golynskiy et al.[69] as previously described[23,25,40] using 0.5 μM Mag-Fura 2 and 1.0 μM apo protein in binding buffer (20 mM 4-(2-hydroxyethyl)-1-piperazineethanesulfonic acid (HEPES) pH 7.2, 200 mM NaCl, 5% glycerol treated with Chelex) containing 1 mM EDTA. Intrinsic fluorescence assays of zinc transfer from PdAztD to apo PdAztC were performed similarly to those previously described[23,25]. PdAztD was reconstituted by addition of 2 molar equivalents of ZnCl₂. The reconstituted protein was then titrated into a solution of 10 μM apo AztC in binding buffer containing 1 mM EDTA. After 1.6 equivalents of AztD were added, ZnCl₂ was added to 1.1 mM to determine saturation. All fluorescence measurements were made using a Varian Cary Eclipse fluorescence spectrophotometer using three independent samples.

**Stopped-flow fluorescence and fluorescence kinetics.** The stopped-flow fluorescence utilized an SX-20 Stopped-Flow apparatus (Applied Photophysics, UK) that was configured for single-wavelength fluorescence detection at 315 nm using a 295 nm LED light source set to 2 amp (Applied Photophysics, UK). A fluorescence PMT detector was tuned to 315 nm via a monochromator, with no inline physical filters therefore necessary. All stopped-flow components were treated exhaustively with EDTA in order to ensure that Zn was delivered only from AztD and not from any contaminating elements. The LED source voltage versus detector signal was calibrated for each experiment by mixing the increasing concentrations of AztD against 20 mM HEPES buffer. Upon reaching peak Tyr/Trp fluorescence signal, the high voltage of the detector was set to ~1 V in order to prevent fluorescence signals from reaching detector saturation as the Zn-loaded AztD to apo AztC metal transfer reactions progressed. Protein concentration was higher overall in the stopped-flow reactions as compared to the static fluorescence work in order to obtain good signal to noise in the fast-mixing conditions. In a typical stopped-flow experiment, Zn-loaded AztD was mixed against 75 μM AztC (mixing concentration 37.5 μM) with increasing concentrations of AztD and the reaction was monitored from the millisecond range to the minute range. Mixing experiments at each concentration were performed twice. Scanning fluorescence was carried out on the stopped flow across the 300–550 nm range to obtain spectra of each reaction before and after mixing experiments. Blank controls were conducted by ensuring that 20 mM HEPES buffer produced no fluorescence by mixing buffer against itself under identical timeframes as that of protein experiments, and AztC was also mixed against buffer to monitor for background fluorescence. Kinetic traces were simulated by GraphPad Prism using a nonlinear (curve) fit for a one-phase association. The fluorescence increase was simulated by the equation $Y = Y0 + (\text{Plateau} - Y0) \times (1 - \exp(-K \times x))$. The relationship between $k_{obs}$ versus [AztD] was fit by linear regression.

Other kinetic experiments were performed using a Varian Cary Eclipse fluorescence spectrophotometer. Apo AztC at 0.5 μM in binding buffer containing 1 mM EDTA was placed in a stirred cell at ambient temperature. Transfer was initiated by addition of varying concentrations of AztD reconstituted as above and fluorescence emission at 315 nm ($\lambda_{exc} = 278$ nm) was recorded over time. Slit widths were varied to optimize signal to noise and avoid saturation at high protein concentration. First order fits to the data were generated using the onboard Cary Eclipse software.

**Metal transfer by anion exchange chromatography.** A HiTrap Q HP anion exchange column (GE Healthcare) was prepared by washing with five column

volumes (CV) of 100 mM EDTA pH 8.0 followed by equilibration with 10 CV of Chelex-treated 20 mM Tris pH 8.0. This same washing protocol was performed in between each sample run. WT or mutant PdAztD was reconstituted as above and combined with apo PdAztC in Chelex-treated 20 mM Tris pH 8.0 with EDTA to generate a final concentration of 100 μM each protein, 200 μM ZnCl₂ (from AztD reconstitution), 1 mM EDTA. The mixture was allowed to incubate at room temperature for 30 min prior to loading onto the column and elution on a gradient of increasing NaCl. Control experiments were performed identically with the absence of either AztC or AztD. Sodium dodecyl sulfate polyacrylamide gel electrophoresis was used to confirm the identity of protein found in each chromatographic fraction. Chromatographic peaks were converted to protein concentration using the respective extinction coefficients[23,40], with the minimum absorbance between the two peaks taken as the division between the two proteins. In some cases, a two-point baseline was applied to absorbance data to compensate for drift. Fractions were collected and digested in 1.6 M HNO₃ at 70° for zinc content analysis by ICP–OES.

**Statistics and reproducibility.** Mag-Fura 2 assays were performed on three independent samples. Each titration was fitted using DynaFit[70,71] to determine the Kd. The average values and standard deviations are reported in Table 2. Kinetics experiments at each concentration were performed on duplicate samples for stopped flow and in triplicate for standard fluorescence. Kinetic traces were fitted as described above and the average apparent rate constants plotted in Fig. 7 along with standard deviations.

**Reporting summary.** Further information on research design is available in the Nature Research Reporting Summary linked to this article.

## Data availability
The atomic coordinates and structure factors for native CkAztD (6CMK), apo CkAztD (6N01), and native PdAztD (6CK1) have been deposited in the Protein Data Bank, Research Collaboratory for Structural Bioinformatics, Rutgers University, New Brunswick, NJ (http://www.rcsb.org/). The raw data for Figs. 4, 5, and 6 and Supplementary Figs. 3, 6, and 8 can be found in Supplementary Information in the Source Data file. All other data supporting the findings of this study can be found in Supplementary Information files and from the corresponding author upon reasonable request.

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

## Acknowledgements

Research reported in this publication was supported by the National Institute of General Medical Science of the National Institutes of Health under award number 1R01GM122819-01A1. We acknowledge the staff at the Berkeley Center for Structural Biology at Lawrence Berkeley National Laboratory. The Berkeley Center for Structural Biology is supported in part by the National Institutes of Health, National Institute of General Medical Sciences, and the Howard Hughes Medical Institute. The Advanced Light Source is supported by the Director, Office of Science, Office of Basic Energy Sciences, of the US Department of Energy under Contract No. DE-AC02-05CH11231. Finally, we acknowledge Hridindu Roychowdhury for tireless efforts in the initial crystallization stages.

## Author contributions

D.P.N. expressed and purified proteins and performed zinc binding and transfer experiments. S.F. expressed and purified proteins and set up crystallization experiments. K.C. performed stopped flow data collection and analysis. E.T.Y. built and refined crystallographic models, devised the experiments, and wrote the paper.

## Additional information

**Competing interests:** The authors declare no competing interests.

