## [Peer Review File · Communications Biology]

Reviewers' comments:

Reviewer #1 (Remarks to the Author):

In this manuscript, the authors report the crystallographic structures of the metallochaperone AztD from *P. denitrificans* (only Zn bound) and *C. koseri* (Zn-bound and apo-state), these are the firsts protein structures from this family of metallochaperones. The authors build on their previously published data on AztD and Zn transporters from *P. denitrificans* and present here a detailed structural and mechanistic study that identifies unequivocally the metal binding site responsible for metal transfer to AztC. Moreover, they propose (based on a phylogenetic analysis) AztD homologues are present in various bacterial taxa, including AztD from *C. koseri* species that they crystallize. Despite the low sequence similarities, the overall topology and the metal binding residues are conserved (here H408 may well be an exception). They validate the model that results from the Zn binding constants (only site 2 AztD is capable of transferring Zn to AztC) by building AztD mutants that are missing site 1 or site 2 and studying their Zn transfer properties. In order to start to unravel the mechanism of Zn transfer, they move on to kinetic experiments where they take advantage of the significant change in intrinsic fluorescence of AztC upon metalation to follow the product (holo-AztC) concentration as a function of time. The experiment is properly performed under pseudo-order conditions using stop-flow. Then, they do some follow up experiments where they do not follow pseudo-order conditions, they obtain some results that are incompatible and it is really hard to follow their interpretation of what seems to be a not well-designed experiment. Finally, they do some docking with their structures of AztD and AztC and the resulting complex is compatible with some of their previous observation on AztC fluorescence and mutagenesis.

This is a novel set of experiments on a system that provides significant insights into the structural biology of Zn(II)-chaperones that were initially discovered by the authors. The subject of Zn chaperones remains controversial given the relatively high availability of Zn(II) in the cell and the lack of detail studies such as the one presented here. This manuscript represents a significant contribution to the structural understanding of Zn uptake, that is obviously not restricted to ABC transporters that have been extensively characterized. Overall the manuscript is well written and easy to follow. The structural data is compelling, as well as their Zn transfer experiments and the docking model is an interesting speculation. However, there are some problems with the experimental design of some of their kinetics experiments (see below). I find the quality of the work is adequate for publication after the authors review the results of the kinetic experiments. There are some key points where I would encourage the authors to give more in-depth discussion.

1) The introduction may be too succinct, it may be valuable for the reader to expand on:

a. The difference between Zn uptake in Gram-negative and Gram-positive bacteria may be useful to understand the significance of metallochaperones in each case. This is relevant since AztD homologues are probably serving different purposes in Gram-negative and Gram-positive bacteria.

For example, in *S. pneumoniae* a Gram-positive bacterium, ZinT is linked to ZnuA, while in most Gram-negative bacteria is thought to be freely diffusing in the periplasms. In this sense, it is interesting to note that *Cmas* AztD is not linked to AztC nor a transmembrane protein, thus this protein must remain associated to the membrane by a different strategy.

b. The data on the biological role of AztD in *P. denitrificans* would be probably much clearer if the Zur regulation and the presence of ZnuABC in that organism are stated in the introduction. Also, it should be mentioned that ZinT is Zur regulated in most bacteria which provides more evidence to compare AztD and ZinT in terms of their biological function.

2) It is exciting to note that AztD homologues are found in a large number of diverse bacterial species (particularly in antibiotic-resistant bacteria) and the metal binding residues identified here by crystallography are conserved.

However, there are many questions remain unanswered in terms of the role of the AztD homologues in the different organisms. What is the genomic context of these AztD-like proteins? Is there any evidence of Zur regulation (Zur-box)? In a previous publication, the authors indicated that the genomic context PdAztD was very similar to the homologues *Mycobacterium* sp., *S. coelicolor* and *N. Farcinica*. Is that general to most of the homologues that they find here (at least the representative that they choose in the sequence alignment)?

The information on metal specificity is so far apparently restricted to demonstrating that the protein binds Zn(II) much tighter than Mn(II). Do the authors expect all the AztD-homologues to be involved in Zn uptake?

Do the organisms that present AztD generally have also a ZnuABC or is *P. denitrificans* an exception? Are there organisms where AztABCD is the single Zn(II) uptake system?

3) The quality of the structures particularly in the metal binding site is very good. The discussion of the holo structures of CkAztC and PdAztD is adequate. Is H408 conserved? It does not seem to be the case in figure 1.

The differences between Apo and Holo CkAztC is summarized only lack electron density of site 2, which the authors interpret as changes in flexibility. Then they indicate that there are not major structural changes, which is very hard to asses in the few residues that are represented in Fig. 3. I would suggest that the CkAztC are presented in Fig. 3 in the same way that PdAztD is on Fig2 (including the order of site 1 and site 2), so the reader can see the structural conservation between apo and Zn CkAztC and compare the overall fold with figure 2 (Fig. S2 is useful for that purpose also). Can the authors expand on the differences?

In this section there are some details that I would suggest the authors include or word differently:

- Add the sequence similarity between CkAztD and PdAztD (page 5).
- Can the authors provide a sequence alignment between AztDs and other metal-binding beta-propeller domain proteins with a different ligand specificity? This can be commented on page 5-6 when the Supplemental file 1 is introduced.
- The caption of Figure 2 can refer to Table S3 when mentioning the anomalous density.
- The paragraph about the occupancies is not easy to follow, the authors may consider narrowing their conclusions to what the crystal structure show rather than speculating on cooperativities that are not measured.

4) The Zn binding and transfer to AztC experiments on the Δ S2 and Δ S1 AztD mutants are successfully validating the interpretation of the structure and the conservation of the residues and identify unequivocally site 2 as functional for Zn transfer. This section is concise and the experiments are well performed. However, reported affinities of AztC should be added to Table 3 and the prediction that one could make based on these affinities (Zn is bound too tightly to site 1 of AztD to ensure transfer to AztC) should be discussed prior to presenting the experiments of this section.

5) The title of the section where the authors try to test possible roles of site 1 is a summary of some negative results based on a possible interpretation of the crystal structure that ends up being crystal packing artifact. I would recommend changing the title of this section since the transfer between site 1 and site 2 does not occur. Are the authors considering the differences in affinities when designing this experiment? It seems that the affinities suggest that there should be an excess of 10-fold to get a significant transfer from site 1 to site 2. Can the authors determine (maybe using ICP-OES) if in Figure S5 if the Zn is still bound to AztD site 1?

It is interesting to note that EDTA can successfully remove Zn from this site and it is not kinetically trapped, while that seems to be the case in AztC (Figure 5). This is probably the most valuable result of this section.

6) The stopped-flow results of this section seem to be well performed. However, a 3.5-fold excess of AztD (150 μ M AztD with 37.5 μ M AztC upon mixing in Fig. 3B) is smaller than one would expect to ensure pseudo-first order condition (for this analysis the concentration of apo-AztD is supposed to remain constant). I would recommend they perform more experiments either at higher AztD concentration or at a lower concentration of AztC. This data clearly indicates a slow transfer that follows first order with respect to holo-AztD and apo-AztC, the global order is 2. Contrary to what the authors indicate in this section, the linear dependence of the single exponential kobs in pseudo-order conditions indicates global order 2.

It is not clear why the authors decide to lower the concentrations to 1 μ M apo-AztC, probably to set up the experiments without the need for a stop-flow apparatus. Here the treatment of pseudo-first order is simply not correct since holo-AztD is not constant. So even if a single exponential curve can successfully fit the data, the linear fit of kobs as a function of [holo AztD] does not have the physical meaning that the authors claim it has.

This misinterpretation of the data may well explain the odd dependence obtained in Figure S6 and the inconsistencies in the second order rate constant. The analysis of the data needs to be performed differently considering how the concentration of each reactant varies as a function of time or the experimental design needs to adjust better to the assumptions of pseudo-first order conditions.

7) The modeling using HADDOCK is very speculative and could be moved to the discussion. It is not clear how 13 Å (from the ab initio modeling) is significantly worse than the 10 Å obtained once they manually dock the structures. Maybe to strengthen the point they could show the structure from ZnuA-ZinT in parallel to the proposed AztC-AztD complex.

Reviewer #2 (Remarks to the Author):

In this very well written manuscript, Yukl and colleagues carry forward their detailed characterization of the *P. denitrificans* *aztABCD* operon to the logical next step. Through a series of high quality crystal structures and well-designed experiments, they contribute new and important details regarding the structure and function of perhaps the most interesting, unique and broadly interesting element of the system: the Zn-binding periplasmic AztD and the mechanism of zinc transfer from AztD to AztC. Most importantly, the authors unambiguously identify site 2 as being the site of zinc transfer, providing a clear new mechanistic understanding of this system to complement the new structural data. The quality of the work is high, the length is appropriate and the impact is highly suitable for publication in *Communications Biology*. Nevertheless, there are some areas that would benefit from further clarification. These are detailed below.

1. Given the size, wide-spread distribution and sequence divergence within the AztD homologues, it would be informative to present a sequence similarity network that illustrates how these proteins are clustered relative to one another. For example, do AztD homologues that are located in an operon with a zinc ABC transporter system cluster separately from those that are not? What other functionally characterized beta-propeller metalloproteins, if any, cluster with this group at higher E values? The Yukl lab has published several excellent prior papers describing the function of AztD but these have not included a thorough description of the sequence relationships within the AztD protein family. Given the phylogenetic analysis presented in the current manuscript and the relative ease with which sequence similarity networks can now be generated, this manuscript seems an appropriate place to present such an analysis. I highly recommend, but won't insist on, the inclusion of an SSN in the current manuscript.
2. Figure S3 shows significant differences in the occupancy and the quality of the 2Fo-Fc maps between sites 2A and 2B. This appears to be due to crystal packing, where site 2A is better ordered by virtue of its interactions with the adjacent monomer (site 1B) in the ASU. This fact is not mentioned in the text on pages 6-7, where the differences in occupancy and density of site 2 are discussed. To what degree are the observed differences in site 2 a result of crystal packing?
3. Figure 4. Should these lines be forced through the origin? If the assumption is that the change in intensity results from zinc transfer, it seems that it should be 0 change in intensity at 0 μM AztD? More importantly, if the "much smaller increases" in fluorescence intensity are "due to the added fluorescence of AztD without Zn transfer", shouldn't a titration of apo AztD in the absence of zinc lead to \sim the same slope changes (~ 10 instead of ~ 20)? It is curious that this control experiment is not provided.
4. Page 9, "These data are consistent with fluorescence results indicating that zinc is transferred only from site 2 while further demonstrating that the zinc at site 1 is labile." This is a very nice and convincing assay but it leaves an unanswered question. Is the zinc at site 1 labile **only in the presence of AztC**? How much faster is Zn lost to solution from AztD in the presence of AztC as compared to in its absence?

5. The experimental design described on page 10 “Zinc transfer from site 1 to site 2 of AztD” raises a question. As Figure 5 demonstrates, the zinc in $\Delta S2$ AztD is labile in the presence of AztC. Perhaps, in this experimental setup, the $\Delta S2$ AztD-AztC interaction competes with the $\Delta S2$ AztD - $\Delta S1$ AztD interaction, releasing the Zn from $\Delta S2$ AztD before it has opportunity to transfer to $\Delta S1$ AztD. If the concentration of $\Delta S1$ AztD was raised to 50 – 100 μ M, would the results be different?
6. Could the “puzzling observation” of reduced k_{on} at elevated [AztD] be explained by concentration-dependent, transient AztD dimerization (similar to the interactions observed in the crystal structures). It is not clear what the authors mean through their rather generic “some autoinhibition” explanation. Also, please note that in the legend for Figure S6, this should be k_{on} and not K_{on} .
7. The docking model is reasonable and sufficient. It will be interesting to further probe this hypothesis through mutagenesis and functional studies, but it would be unreasonable for any reviewer to ask this to be inserted into the current manuscript. That said, there is arguably an excessive reliance on this docking model in the third paragraph of the discussion, considering that much work yet needs to be performed in order to confirm this model.

Reviewers' comments:

Reviewer #1 (Remarks to the Author):

In this manuscript, the authors report the crystallographic structures of the metallochaperone AztD from *P. denitrificans* (only Zn bound) and *C. koseri* (Zn-bound and apo-state), these are the firsts protein structures from this family of metallochaperones. The authors build on their previously published data on AztD and Zn transporters from *P. denitrificans* and present here a detailed structural and mechanistic study that identifies unequivocally the metal binding site responsible for metal transfer to AztC. Moreover, they propose (based on a phylogenetic analysis) AztD homologues are present in various bacterial taxa, including AztD from *C. koseri* species that they crystallize. Despite the low sequence similarities, the overall topology and the metal binding residues are conserved (here H408 may well be an exception). They validate the model that results from the Zn binding constants (only site 2 AztD is capable of transferring Zn to AztC) by building AztD mutants that are missing site 1 or site 2 and studying their Zn transfer properties. In order to start to unravel the mechanism of Zn transfer, they move on to kinetic experiments where they take advantage of the significant change in intrinsic fluorescence of AztC upon metalation to follow the product (holo-AztC) concentration as a function of time. The experiment is properly performed under pseudo-order conditions using stop-flow. Then, they do some follow up experiments where they do not follow pseudo-order conditions, they obtain some results that are incompatible and it is really hard to follow their interpretation of what seems to be a not well-designed experiment. Finally, they do some docking with their structures of AztD and AztC and the resulting complex is compatible with some of their previous observation on AztC fluorescence and mutagenesis.

This is a novel set of experiments on a system that provides significant insights into the structural biology of Zn(II)-chaperones that were initially discovered by the authors. The subject of Zn chaperones remains controversial given the relatively high availability of Zn(II) in the cell and the lack of detail studies such as the one presented here. This manuscript represents a significant contribution to the structural understanding of Zn uptake, that is obviously not restricted to ABC transporters that have been extensively characterized. Overall the manuscript is well written and easy to follow. The structural data is compelling, as well as their Zn transfer experiments and the docking model is an interesting speculation. However, there are some problems with the experimental design of some of their kinetics experiments (see below). I find the quality of the work is adequate for publication after the authors review the results of the kinetic experiments. There are some key points where I would encourage the authors to give more in-depth discussion.

- 1) The introduction may be too succinct, it may be valuable for the reader to expand on:
 - a. The difference between Zn uptake in Gram-negative and Gram-positive bacteria may be useful to understand the significance of metallochaperones in each case. This is relevant since AztD homologues are probably serving different purposes in Gram-negative and Gram-positive bacteria. For example, in *S. pneumoniae* a Gram-positive bacterium, ZinT is linked to ZnuA, while in most Gram-negative bacteria is thought to be freely diffusing in the periplasms. In this sense, it is interesting to note that *Cmas* AztD is not linked to AztC nor a transmembrane protein, thus this protein must remain associated to the membrane by a different strategy.
 - b. The data on the biological role of AztD in *P. denitrificans* would be probably much clearer if the Zur regulation and the presence of ZnuABC in that organism are stated in the introduction. Also, it should be mentioned that ZinT is Zur regulated in most bacteria which provides more evidence to compare AztD and ZinT in terms of their biological function.

The introduction and first paragraph of the results have been revised and expanded to address these issues.

2) It is exciting to note that AztD homologues are found in a large number of diverse bacterial species (particularly in antibiotic-resistant bacteria) and the metal binding residues identified here by crystallography are conserved. However, there are many questions remain unanswered in terms of the role of the AztD homologues in the different organisms. What is the genomic context of these AztD-like proteins? Is there any evidence of Zur regulation (Zur-box)? In a previous publication, the authors indicated that the genomic context PdAztD was very similar to the homologues *Mycobacterium* sp., *S. coelicolor* and *N. Farcinica*. Is that general to most of the homologues that they find here (at least the representative that they choose in the sequence alignment)?

The information on metal specificity is so far apparently restricted to demonstrating that the protein binds Zn(II) much tighter than Mn(II). Do the authors expect all the AztD-homologues to be involved in Zn uptake?

Do the organisms that present AztD generally have also a ZnuABC or is *P. denitrificans* an exception? Are there organisms where AztABCD is the single Zn(II) uptake system?

We have revised and expanded the first section of Results to include a sequence similarity network and genome neighborhood network (Fig 1, sequence alignment has been moved to S1). The results for the distribution of AztD homologues among different taxa are slightly different, owing to the fact that a different database (UniProtKB) was interrogated. 577 species are identified with AztD homologues and these cluster very nicely along phylogenetic lines. Across all sequences, the predominant genome neighborhood features would suggest involvement of AztD in zinc import and homeostasis. However, in some clusters it may have divergent functions (see cluster 4 in Figure S2). We have included the complete group of genome neighborhood diagrams as supplemental file 2.

The last two questions are difficult to address. The cluster A-I SBPs are similar to one another. This includes AztC and ZnuA as well as the Mn-transporting SBPs. Thus, we are uncertain how to address this problem without manually searching individual genomes for the presence of multiple cluster A-I SBPs. We can say that *C. koseri* also has a ZnuABC system but a random search of other genomes quickly turned up a species that does not appear to have ZnuABC (e.g. *Brevibacterium aurantiacum*). Thus, the appearance of redundant Zn ABC transporter systems is likely to be variable.

3) The quality of the structures particularly in the metal binding site is very good. The discussion of the holo structures of CkAztC and PdAztD is adequate. Is H408 conserved? It does not seem to be the case in figure 1.

The differences between Apo and Holo CkAztC is summarized only lack electron density of site 2, which the authors interpret as changes in flexibility. Then they indicate that there are not major structural changes, which is very hard to assess in the few residues that are represented in Fig. 3. I would suggest that the CkAztC are presented in Fig. 3 in the same way that PdAztD is on Fig2 (including the order of site 1 and site 2), so the reader can see the structural conservation between apo and Zn CkAztC and compare the overall fold with figure 2 (Fig. S2 is useful for that purpose also). Can the authors expand on the differences?

H408 is conserved in only 68.1% of analyzed AztD sequences. This is now included on p. 7. Figure 3 and the text describing it has been modified as suggested.

In this section there are some details that I would suggest the authors include or word differently:

- Add the sequence similarity between CkAztD and PdAztD (page 5).
- Can the authors provide a sequence alignment between AztDs and other metal-binding beta-propeller domain proteins with a different ligand specificity? This can be commented on page 5-6 when the Supplemental file 1 is introduced.
- The caption of Figure 2 can refer to Table S3 when mentioning the anomalous density.
- The paragraph about the occupancies is not easy to follow, the authors may consider narrowing their

conclusions to what the crystal structure show rather than speculating on cooperativities that are not measured.

- The sequence ID and similarity between Ck and PdAztD is now included on p. 5.
- The very low sequence identity between AztD and the other known metal binding beta propellers make alignments between them uninformative. Rather, the following text has been added to p. 6: "While NiR and N₂OR bind heme and CuZ cluster, respectively, they have very low sequence identity to PdAztD (12.3% and 11.4%) and metal binding residues are not conserved. Further, neither is known to participate in any type of metal transfer. Thus, AztD is the first beta propeller protein to our knowledge to have a role in metal transfer, highlighting and expanding the diversity of functions for which this fold has evolved."
- A reference to Table S3 has been added to Fig. 2 caption.
- The discussion of cooperative binding has been removed from this section.

4) The Zn binding and transfer to AztC experiments on the Δ S2 and Δ S1 AztD mutants are successfully validating the interpretation of the structure and the conservation of the residues and identify unequivocally site 2 as functional for Zn transfer. This section is concise and the experiments are well performed. However, reported affinities of AztC should be added to Table 3 and the prediction that one could make based on these affinities (Zn is bound too tightly to site 1 of AztD to ensure transfer to AztC) should be discussed prior to presenting the experiments of this section.

AztC affinity has been added to table 3. The affinities are actually comparable between WT AztD, mutant AztD and AztC. This led us to previously propose that transfer is under kinetic rather than thermodynamic control (see *J. Biol. Chem.* 290, 29984-92). A statement to that effect has been added to p.8.

5) The title of the section where the authors try to test possible roles of site 1 is a summary of some negative results based on a possible interpretation of the crystal structure that ends up being crystal packing artifact. I would recommend changing the title of this section since the transfer between site 1 and site 2 does not occur. Are the authors considering the differences in affinities when designing this experiment? It seems that the affinities suggest that there should be an excess of 10-fold to get a significant transfer from site 1 to site 2. Can the authors determine (maybe using ICP-OES) if in Figure S5 if the Zn is still bound to AztD site 1?

It is interesting to note that EDTA can successfully remove Zn from this site and it is not kinetically trapped, while that seems to be the case in AztC (Figure 5). This is probably the most valuable result of this section.

The title of this section has been changed to "Zinc is not transferred from site 1 to site 2 of AztD". It would be difficult to measure the transfer in very high concentrations of DS1 as the background fluorescence would be very high. Similarly, DS1 and DS2 would co-elute during IEC, making it impossible to differentiate which held the Zn by ICP. We have included a control experiment of reconstituted AztD alone incubated in the presence of EDTA for 30 min prior to ion exchange chromatography as used to determine transfer (Fig. 5E). This experiment demonstrates that both zinc sites are labile even in the absence of AztC. Thus, the purpose of the experiment was really to determine whether the hybrid zinc site was physiologically meaningful and if transfer between AztD sites is kinetically controlled and can compete with simple dissociation. This does not appear to be the case and the text in this section has been modified accordingly.

6) The stopped-flow results of this section seem to be well performed. However, a 3.5-fold excess of AztD (150 μ M AztD with 37.5 μ M AztC upon mixing in Fig. 3B) is smaller than one would expect to ensure pseudo-first order condition (for this analysis the concentration of apo-AztD is supposed to remain constant). I would recommend they perform more experiments either at higher AztD concentration or at a

lower concentration of AztC. This data clearly indicates a slow transfer that follows first order with respect to holo-AztD and apo-AztC, the global order is 2. Contrary to what the authors indicate in this section, the linear dependence of the single exponential kobs in pseudo-order conditions indicates global order 2. It is not clear why the authors decide to lower the concentrations to 1uM apo-AztC, probably to set up the experiments without the need for a stop-flow apparatus. Here the treatment of pseudo-first order is simply not correct since holo-AztD is not constant. So even if a single exponential curve can successfully fit the data, the linear fit of kobs as a function of [holo AztD] does not have the physical meaning that the authors claim it has.

This misinterpretation of the data may well explain the odd dependence obtained in Figure S6 and the inconsistencies in the second order rate constant. The analysis of the data needs to be performed differently considering how the concentration of each reactant varies as a function of time or the experimental design needs to adjust better to the assumptions of pseudo-first order conditions.

We acknowledge that a 4-fold excess is at the low end of what might be considered pseudo first order. We are limited in the range of AztC:AztD ratios we can use because of the added fluorescence intensity from excess AztD, which decreases signal to noise at high concentrations. However, there is precedent for using similar ratios for this purpose in the literature (see *J. Biol. Chem.* 282, 31380–8, *J. Biol. Chem.* 288, 25183–93, and *Biochemistry* 49, 6646-64). Further, the nearly perfect linear fit of k_{obs} vs. [AztD] in the stopped-flow data suggests that we are in the pseudo-first order regime. We have changed “bimolecular rate constant” to “second order rate constant for the overall process” at the end of p. 10 to clarify that the global order is indeed 2.

The treatment of the non-stopped-flow kinetic data as pseudo-first order (Fig. 6D) is admittedly incorrect. We have repeated these experiments under pseudo-first order conditions with $[AztD] \geq 5x[apoAztC]$ and replaced Fig. 6C and 6D with this data. The description of the kinetic data in Results has been simplified and Figure S6 has been removed. Additional interpretation of kinetic data has been moved to the discussion along with the description of the docking model (see response to point 7).

7) The modeling using HADDOCK is very speculative and could be moved to the discussion. It is not clear how 13 Å (from the ab initio modeling) is significantly worse than the 10 Å obtained once they manually dock the structures. Maybe to strengthen the point they could show the structure from ZnuA-ZinT in parallel to the proposed AztC-AztD complex.

We have moved the model and the more speculative elements of the kinetics analysis to the Discussion. The model of the ZinT-ZnuA complex is derived from low resolution SAXS data and requires some interpretation. Thus, we would prefer to avoid reproducing such a structure and stay with the authors' original interpretation.

Reviewer #2 (Remarks to the Author):

In this very well written manuscript, Yuki and colleagues carry forward their detailed characterization of the *P. denitrificans* aztABCD operon to the logical next step. Through a series of high quality crystal structures and well-designed experiments, they contribute new and important details regarding the structure and function of perhaps the most interesting, unique and broadly interesting element of the system: the Zn-binding periplasmic AztD and the mechanism of zinc transfer from AztD to AztC. Most importantly, the authors unambiguously identify site 2 as being the site of zinc transfer, providing a clear new mechanistic understanding of this system to complement the new structural data. The quality of the work is high, the length is appropriate and the impact is highly suitable for publication in *Communications Biology*. Nevertheless, there are some areas that would benefit from further clarification. These are detailed below.

1. Given the size, wide-spread distribution and sequence divergence within the AztD homologues, it would be informative to present a sequence similarity network that illustrates how these proteins are clustered relative to one another. For example, do AztD homologues that are located in an operon with a zinc ABC transporter system cluster separately from those that are not? What other functionally characterized beta-propeller metalloproteins, if any, cluster with this group at higher E values? The Yukl lab has published several excellent prior papers describing the function of AztD but these have not included a thorough description of the sequence relationships within the AztD protein family. Given the phylogenetic analysis presented in the current manuscript and the relative ease with which sequence similarity networks can now be generated, this manuscript seems an appropriate place to present such an analysis. I highly recommend, but won't insist on, the inclusion of an SSN in the current manuscript.

An SSN and genome neighborhood network have been included in the revised manuscript as Fig. 1. Fig. S2 shows genome neighborhood networks for the 5 largest clusters of the SSN. While Zn transport and homeostasis genes are the most consistently associated with AztD, there is at least one cluster where the pattern is significantly different. To our knowledge, only nitrite reductase and nitrous oxide reductase have beta propeller domains that bind metals. These exhibit very low sequence identity to AztD and were not included in the SSN, although they are now briefly discussed on p. 6 of the revised manuscript.

2. Figure S3 shows significant differences in the occupancy and the quality of the 2Fo-Fc maps between sites 2A and 2B. This appears to be due to crystal packing, where site 2A is better ordered by virtue of its interactions with the adjacent monomer (site 1B) in the ASU. This fact is not mentioned in the text on pages 6-7, where the differences in occupancy and density of site 2 are discussed. To what degree are the observed differences in site 2 a result of crystal packing?

The fact that differences in occupancy may be due to crystal packing is now addressed on p. 6. The fact that site 2 residues are more disordered in the apo *CkAztD* structure is now addressed at this point and shown in the revised Fig. 3. While crystal packing surely has an impact on occupancy and order at site 2, the absence of zinc clearly leads to disorder at this site.

3. Figure 4. Should these lines be forced through the origin? If the assumption is that the change in intensity results from zinc transfer, it seems that it should be 0 change in intensity at 0 μM AztD? More importantly, if the "much smaller increases" in fluorescence intensity are "due to the added fluorescence of AztD without Zn transfer", shouldn't a titration of apo AztD in the absence of zinc lead to \sim the same slope changes (~ 10 instead of ~ 20)? It is curious that this control experiment is not provided.

Fig. 4 has been revised such that fit lines are forced through the origin. The control experiment was performed in a previous publication (see Fig. 7B in *J. Biol. Chem.* 290, 29984-92) and indeed generates a slope of 10.6. This is now explicitly mentioned on p. 8.

4. Page 9, "These data are consistent with fluorescence results indicating that zinc is transferred only from site 2 while further demonstrating that the zinc at site 1 is labile." This is a very nice and convincing assay but it leaves an unanswered question. Is the zinc at site 1 labile only in the presence of AztC? How much faster is Zn lost to solution from AztD

in the presence of AztC as compared to in its absence?

We have performed the control experiment of reconstituted AztD alone treated as the other samples in this assay (Fig. 5E). Somewhat surprisingly, both zinc ions are lost. The last sentence of this paragraph has been amended as follows:

“These data are consistent with fluorescence results indicating that zinc is transferred only from site 2. Finally, reconstituted AztD in the absence of AztC loses both zinc ions during incubation with EDTA and subsequent chromatography, demonstrating that both AztD zinc sites are relatively labile. Thus, AztC acts as a kinetic trap for zinc transferred from AztD in the presence of competing chelators.

Although it will be intriguing to more precisely measure off-rates for zinc from AztD, these experiments are not trivial and perhaps beyond the scope of the current manuscript. However, we can say that transfer from site 2 of AztD to AztC must be more rapid than its off-rate.

5. The experimental design described on page 10 “Zinc transfer from site 1 to site 2 of AztD” raises a question. As Figure 5 demonstrates, the zinc in DS2 AztD is labile in the presence of AztC. Perhaps, in this experimental setup, the DS2AztD-AztC interaction competes with the DS2 AztD - DS1 AztD interaction, releasing the Zn from DS2 AztD before it has opportunity to transfer to DS1 AztD. If the concentration of DS1 AztD was raised to 50 – 100 μM , would the results be different?

It would be difficult to measure the transfer in very high concentrations of DS1 as the background fluorescence would be very high. The above observations demonstrate that both zinc sites are labile even in the absence of AztC. Thus, the purpose of the experiment was really to determine whether the hybrid zinc site was physiologically meaningful and if transfer between AztD sites is kinetically controlled and can compete with simple dissociation. This does not appear to be the case and the text in this section has been modified accordingly.

6. Could the “puzzling observation” of reduced k_{on} at elevated [AztD] be explained by concentration-dependent, transient AztD dimerization (similar to the interactions observed in the crystal structures). It is not clear what the authors mean through their rather generic “some autoinhibition” explanation. Also, please note that in the legend for Figure S6, this should be k_{on} and not K_{on} .

The non-stopped-flow kinetics experiments have been repeated using true pseudo-first order conditions. The previous analysis of this data was flawed and the discussion of autoinhibition and figure S6 have been deleted.

7. The docking model is reasonable and sufficient. It will be interesting to further probe this hypothesis through mutagenesis and functional studies, but it would be unreasonable for any reviewer to ask this to be inserted into the current manuscript. That said, there is arguably an excessive reliance on this docking model in the third paragraph of the discussion, considering that much work yet needs to be performed in order to confirm this model.

The docking model has been moved to the Discussion section to emphasize the speculative nature of this section.

REVIEWERS' COMMENTS:

Reviewer #1 (Remarks to the Author):

The authors have adequately addressed all of my concerns. The revised manuscript is much better than the prior version and of sufficient quality for publication in my opinion.

Reviewer #2 (Remarks to the Author):

The authors have appropriately responded to all of my suggestions and critiques